# Evaluation and Validation on Sensitivity of Near-Infrared Diffuse Reflectance in Non-Invasive Human Blood Glucose Measurement

**DOI:** 10.3390/s24185879

**Published:** 2024-09-10

**Authors:** Qing Ge, Tongshuai Han, Rong Liu, Zengfu Zhang, Di Sun, Jin Liu, Kexin Xu

**Affiliations:** 1State Key Laboratory of Precision Measurement Technology and Instruments, Tianjin University, Tianjin 300072, China; 2Sunrise Technology Co., Ltd., Tianjin 300192, China

**Keywords:** non-invasive glucose measurement, blood glucose, near-infrared spectroscopy, sensitivity

## Abstract

In non-invasive blood glucose measurement, the sensitivity of glucose-induced optical signals within human tissue is a crucial reference point. This study evaluates the sensitivity of glucose-induced diffuse reflectance in the 1000–1700 nm range. A key factor in understanding this sensitivity is the rate at which the scattering coefficient changes due to glucose, as it is significantly higher than in non-living media and predominantly influences the diffuse light signal level when blood glucose levels change. The study measured and calculated the changes in the scattering coefficient at 1314 nm, a wavelength chosen for its minimal interference from glucose absorption and other bodily constituents. Based on the Mie scattering theory and the results at 1314 nm, the changes in the scattering coefficient within the 1000–1700 nm range were estimated. Subsequently, the sensitivity of the glucose signal across this range was determined through Monte Carlo (MC) simulations. The findings from 25 human trials indicate that the measured sensitivities at five other typical wavelengths within this band generally align with the sensitivities calculated using the aforementioned method. This research can guide the identification of blood glucose signals and the selection of wavelengths for non-invasive blood glucose measurements.

## 1. Introduction

Non-invasive blood glucose monitoring plays a crucial role in the management of diabetes and patient health. Near-infrared spectroscopy emerges as a promising technique for non-invasive glucose measurement, primarily due to its strong tissue penetration capability [1]. This enables the acquisition of glucose information from both tissue fluid and blood within the dermis layer. The methodology and associated hardware systems have relatively matured. However, similar to other optical measurement techniques, in vivo glucose signals captured through near-infrared light are susceptible to interference from physiological variations and other components present in tissues [2,3]. The human body, being a dynamic biological system, presents intricate physiological processes and regulatory mechanisms that pose significant challenges in accurately extracting glucose information from optical signals.

An understanding of the magnitude and characteristics of the sensitivity caused by a unit change in blood glucose concentration is necessary. Developers of instruments are required to design the hardware resolution of the measurement system based on this sensitivity, as well as select the wavelengths of light and the source–detector separations (SDSs) to be used and develop appropriate signal processing approaches.

As directly obtaining the signal of glucose for living tissues is typically challenging, most research groups combine near-infrared diffuse transmittance/reflection spectroscopy with chemometric methods to extract in vivo blood glucose information [4,5,6]. These approaches largely rely on data correlation or projection [7], extracting glucose information from a statistical standpoint without quantifying glucose sensitivity based on physical or chemical principles.

Only a few groups focus on the direct sensing of glucose under discrete single wavelengths. Han et al. of our research team has made significant progress in in vivo signal acquisition strategy both in sensors and differential data processing, and they have succeeded in observing glucose signals directly [8]. They measured the glucose in human arm skin at six wavelengths (within the 1000–1700 nm range) using differentiating processing on the diffuse reflectance of two SDSs to eliminate common-mode interference. However, they still cannot observe the glucose signals through the diffuse reflectance at the singular SDS. In this paper, we draw on Han’s method and utilize differential measurement techniques to estimate the changes in skin scattering coefficients caused by glucose. Then, we indirectly calculate the glucose sensitivity under a different single SDS. This differential measurement approach aids in obtaining reliable in vivo measurement results. The glucose signals derived from differential measurements exhibit distinct magnitudes and spectral characteristics compared to those obtained at a single SDS. Only by understanding the sensitivity characteristics at different SDSs can we flexibly choose the method and corresponding SDSs according to the measurement needs.

Many studies have employed simulation techniques, particularly the MC method, to identify the magnitude and characteristics of the sensitivity of glucose signals in human tissues. This approach has seen rapid development in recent years and is now capable of simulating the absorption and scattering processes of photons in highly complex human tissue structures [9,10]. However, this modeling and simulation approach requires the determination of the change of both absorption and scattering coefficients in living tissues in response to blood glucose variations. Changes in glucose lead to corresponding changes in the tissue’s absorption and scattering characteristics. Among these effects, the absorption change is relatively more straightforward, mainly due to the distinct absorption properties of glucose molecules at specific NIR wavelengths [11], while the scattering is more complex, being closely associated with the state of the scattering media and susceptible to the tissue structure and composition [12]. Therefore, figuring out the change in the scattering of living tissues caused by glucose is essential for acquiring an accurate evaluation of in vivo glucose signal sensitivity.

Attempts have been made to use biological phantoms or tissue slices as substitutes for live animal tissues to study the scattering characteristics caused by glucose changes [13]. Experiments have shown that the scattering changes induced by glucose in phantoms or slice experiments are noticeably different from those in live animal experiments. In living tissues, the variability of tissue scattering coefficients caused by glucose may be significantly higher than those observed in similar experiments with slice tissues or phantom solutions [14,15,16]. Matti Kinnunen et al. conducted a study comparing the scattering changes caused by glucose in 5% intralipid solution, tissue slices, and live mice. Using 910 nm light, a glucose concentration change of 1 mmol/L resulted in only a −0.07% change in the scattering coefficient in intralipid solution compared to −1.37% and −2.7% changes in sliced skin and live mice, respectively [15]. This suggests that the changes in scattering coefficients induced by glucose in living tissues compared to slice tissues can differ by several times, even up to tenfold.

In obtaining accurate data through in vivo animal tests, Larin et al. utilized Optical Coherence Tomography (OCT) on live animals. They found that with a 1 mmol/L increase in blood glucose concentration and using 1300 nm light, the animal’s living tissue scattering coefficient decreased by 0.22% [12]. However, due to the limited depth of OCT, these results were obtained using wavelengths where tissue absorption is weaker. To date, no studies have verified whether the scattering coefficient changes measured with these wavelengths are applicable to wavelengths with stronger absorption. Therefore, obtaining scattering coefficient variation data in the 1000–1700 nm band is crucial for fully understanding the impact of glucose fluctuations on scattering coefficients. These data will help determine the sensitivity within using a wavelength band, selecting the measuring wavelength to achieve the optimal possible glucose response.

This paper begins by defining the sensitivity of absorbance to glucose changes in in vivo measurements and analyzes the parameters required to estimate this absorbance sensitivity. In light of the challenges in acquiring data on scattering coefficient variations caused by glucose, we introduce a method that integrates MC simulations with in vivo experiments. This method is employed to estimate the alterations in scattering coefficients due to glucose concentration changes when using monochromatic light in in vivo measurements. Furthermore, by applying the Mie scattering theory [17], we extend the observed changes in scattering coefficients with monochromatic light to the 1000–1700 nm wavelength range. Additionally, we conducted MC simulations for a three-layer skin model to acquire the absorbance sensitivity of glucose in this wavelength range. The human OGTTs are conducted with five typical wavelengths in the 1000–1700 nm range to validate the calculated sensitivity, while the method of extending the scattering properties of 1314 nm light to the 1000–1700 nm range is also validated as feasible.

We utilized the high-precision monitoring equipment and in vivo measurement approach described in Han’s paper, ensuring the accuracy and reliability of the human experimental data in our study. Building upon Han’s foundation, we conducted rigorous theoretical calculations to determine the sensitivity of blood glucose at any given wavelength and source–detector separation within the 1000–1700 nm range and then validated these calculations through extensive human experimentation. Our primary focus was to measure the change in the scattering coefficient caused by variations in glucose concentration in human skin, a detail not addressed in Han’s findings. Therefore, our work provides significant theoretical insights relevant to understanding the components of the glucose signal. It enables us to assess sensitivity regardless of whether the signal arises from glucose absorption or scattering, offering valuable data that are crucial for designing spectroscopic measurement instruments.

## 2. Methods and Experiments

### 2.1. Theoretical Calculation of Sensitivity of Glucose

In the glucose measurement using NIR diffuse reflectance spectroscopy, the modified Beer–Lambert’s law is commonly applied to describe the reflected diffuse light intensity [18]:(1)$$I=I_{0}\cdot \exp\;[-\mu_{a}\cdot L\left(\mu_{a},\mu_{s},r,g\right)-G]$$
where I is the diffuse reflectance intensity, and $I_{0}$ is the incident light intensity. L is the equivalent path length, which depends on the tissue’s absorption coefficient $\mu_{a}$, scattering coefficient $\mu_{s}$, anisotropy factor g, and the SDS r. G is a loss factor.

The absorbance at the SDS r is given by:(2)$$A(r)=-ln\frac{I(r)}{I_{0}}$$

Sensitivity $S_{A}$ is defined as the change in absorbance caused by a 1 mmol/L change in glucose concentration, defined as:(3)$$S_{A}=\frac{dA}{dC_{g}}$$

The glucose primarily affects the optical signal by altering the tissue’s absorption coefficient $\mu_{a}$ and scattering coefficient $\mu_{s}$. Thus, the differential form of Equation (3) is expressed as:(4)$$S_{A}=\frac{dA}{dC_{g}}=\frac{\partial A}{\partial \mu_{a}}\cdot \frac{d\mu_{a}}{{dC}_{g}}+\frac{\partial A}{\partial \mu_{s}}\cdot \frac{d\mu_{s}}{{dC}_{g}}$$
where $\frac{\partial A}{\partial \mu_{a}}$ and $\frac{\partial A}{\partial \mu_{s}}$ represent the proportionality factors of the changes in absorbance due to changes in the absorption coefficient and the scattering coefficient. In Equation (4), the first term represents the impact of absorption, denoted as $S_{\mu_{a}}=\frac{\partial A}{\partial \mu_{a}}\cdot \frac{d\mu_{a}}{{dC}_{g}}$, and the second term represents the impact of scattering, denoted as $S_{\mu_{s}}=\frac{\partial A}{\partial \mu_{s}}\cdot \frac{d\mu_{s}}{{dC}_{g}}$. Thus, $S_{A}=S_{\mu_{a}}+S_{\mu_{s}}$. Under normal circumstances, with known optical parameters of the tissue, such as $\mu_{a}$, $\mu_{s}$, anisotropy factor g, and refractive index n, MC simulation can obtain $\frac{\partial A}{\partial \mu_{a}}$ and $\frac{\partial A}{\partial \mu_{s}}$. The proportionality factors of the absorption coefficient due to glucose, $\frac{d\mu_{a}}{{dC}_{g}}$, is commonly believed to be related to the molar extinction coefficient of glucose in aqueous media or phantom solutions. However, as mentioned above, the $\frac{d\mu_{s}}{{dC}_{g}}$ in living tissues shows significant differences from its behavior in aqueous solutions, intralipid solutions, and ex vivo tissue slices and can only be determined through in vivo measurements. Therefore, determining $\frac{d\mu_{s}}{{dC}_{g}}$ in in vivo tissues is key to estimating the sensitivity of the glucose signal.

### 2.2. Monte Carlo Simulation

The tetrahedron-based inhomogeneous Monte Carlo simulator MC eXtreme (MCX), version 1.9.7 (v2022.10, Heroic Hexagon, Boston, MA, USA), proposed by Shen et al. [19], is used in this paper. Simulation parameters are set as follows: The skin is modeled as a three-layer structure, with the epidermis, dermis, and subcutaneous layers having thicknesses of 0.1 mm, 1.0 mm, and 10.0 mm, respectively. The optical parameters for each skin layer are derived from the literature [20]. The probe interface is made of glass with a refractive index of n = 1.4 and a thickness of 0.2 mm. Diffuse reflectance photons are collected at five SDSs of 1.7 mm, 2.0 mm, 2.3 mm, 2.6 mm, and 2.9 mm, with 10^10^ incident photons.

### 2.3. Estimating the Change in Scattering Coefficient Caused by Glucose

When employing light within 1000–1700 nm, the variation in the scattering coefficient caused by glucose can be inferred from that given value at one wavelength in this range. It necessitates understanding the relationship between the wavelengths. According to the Mie scattering theory, within a simplified medium model with spherical scattering particles, the reduced scattering coefficient $\mu_{s}^{\prime }$ is approximately expressed as [17]:(5)$$\mu_{s}^{\prime }=K\cdot {\left(\frac{n_{s}}{n_{0}}-1\right)}^{2}$$

Here, $\mu_{s}^{\prime }=\mu_{s}\cdot (1-g)$ is the reduced scattering coefficient. K is a proportionality factor related to the density and size of the scattering particles and wavelength, with $n_{s}$ and $n_{0}$ representing the refractive indices of the scattering particle and medium, respectively. The relative change in the reduced scattering coefficient with increased glucose concentration can be expressed as:(6)$$\frac{d\mu_{s}^{\prime }/dC_{g}}{\mu_{s}^{\prime }}=\frac{2n_{s}}{n_{0}(n_{0}-n_{s})}\cdot \frac{dn_{0}}{dC_{g}}$$

The term $\frac{2n_{s}}{n_{0}(n_{0}-n_{s})}$ can be considered as a constant in the NIR spectrum, while $\frac{dn_{0}}{{dC}_{g}}$ represents the linear change in the solution’s refractive index with glucose and is also approximated as constant [17,18,21]. The variations in the anisotropy factor g due to glucose also can be ignored. Therefore, it is inferred that the relative change in the scattering coefficient caused by glucose is nearly constant across the 1000–1700 nm range. Then, the relationship between the wavelengths can be easily acquired so that we can estimate $\frac{d\mu_{s}}{{dC}_{g}}$ for all the wavelengths by determining it at one wavelength first.

We selected 1314 nm light to measure $\frac{d\mu_{s}}{{dC}_{g}}$ of human skin, less affected by glucose absorption [22], hemoglobin [23], water content [24], and body temperature fluctuations [25]. The result of 1314 nm will be extended to all wavelengths.

The in vivo measurement strategy with differential method was adopted in the human experiment, referring to the report of Han et al. [8]. The strategy ensures consistent conditions, while the differential processing effectively counters common-mode background interference in in vivo measurements.

The differential absorbance, $A_{D}$, is defined as:(7)$$A_{D}=A\left(r_{A}\right)-A\left(r_{B}\right),r_{A}>r_{B}$$
where $r_{A}$ and $r_{B}$ represent two different SDSs, and $A\left(r_{A}\right)$ and $A\left(r_{B}\right)$ are the absorbance at the two SDSs, respectively. The scattering coefficient variation due to glucose can be represented as:(8)$$\frac{d\mu_{s}}{{dC}_{g}}=\frac{dA_{D}}{dC_{g}}/\frac{dA_{D}}{d\mu_{s}}$$

$\frac{dA_{D}}{dC_{g}}$ signifies the sensitivity of differential absorbance $A_{D}$, and $\frac{dA_{D}}{d\mu_{s}}$ indicates the proportional factors of $A_{D}$ and $\mu_{s}$.

The $\frac{dA_{D}}{d\mu_{s}}$ can be estimated by using MC simulation data. Then, $\frac{d\mu_{s}}{{dC}_{g}}$ can be calculated by using Equation (8). The two SDSs are $r_{A}$ = 1.7 mm and $r_{B}$ = 2.0 mm. The simulated $A\left(r_{A}\right)$ and $A\left(r_{B}\right)$ of 1314 nm with the varying scattering coefficients are shown in Figure 1. The absorption coefficient is 2.05 mm^−1^ [20], and the scattering coefficient $\mu_{s}$ of dermis varies from 5 to 13 mm^−1^. The slopes of the fitting curves in Figure 1a,b can be given as $\frac{dA\left(r_{A}\right)}{d\mu_{s}}$ and $\frac{dA\left(r_{B}\right)}{d\mu_{s}}$. The difference between these slopes, $\frac{dA_{D}}{d\mu_{s}}=\frac{dA\left(r_{A}\right)-dA\left(r_{B}\right)}{d\mu_{s}}$, is 0.0102 a.u.·mm^−1^.

### 2.4. Human Subject Experiments

The primary aim of this subsection is to obtain the relative change in the scattering coefficient with increased glucose concentration through human experiments. To obtain this value, we conducted 25 experiments, limiting the measurement site to the same location on the forearm and involving only Asian subjects. In selecting participants, we also aimed to ensure diversity, with male and female subjects ranging in age from 23 to 75 years. Subjects participating in the OGTT experiment included 6 elderly men (over 65 years old), 7 elderly women (over 65 years old), 6 young men (under 35 years old), and 6 young women (under 35 years old). In the human experiment with 1314 nm light, the $\frac{dA_{D}}{dC_{g}}$ can be acquired from the measured absorbance $A\left(r_{A}\right)$ and $A\left(r_{B}\right)$ with the glucose concentration in OGTT.

The experimental setup comprised a light source array, an optical switch, a sensing probe, and a unit for data acquisition and processing, as shown in Figure 2. The light source array consisted of six superluminescent diodes with central wavelengths of 1050 nm, 1219 nm, 1314 nm, 1380 nm, 1550 nm, and 1609 nm, respectively. The 3 dB bandwidths of the six light sources are, respectively, 51 nm, 32 nm, 36 nm, 58 nm, 52 nm, and 57 nm. The incident light was guided by optical fibers, diffusely reflected by the skin, and then received by the detector. The detector is custom-made, consisting of five concentric rings of InGaAs photosensitive surfaces with different diameters [8]. Each ring independently receives diffusely reflected light from five different SDS locations (1.7 mm, 2.0 mm, 2.3 mm, 2.6 mm, and 2.9 mm). To mitigate the effects of skin temperature heterogeneity and fluctuations, the measurement temperature was set higher than the natural body temperature. The skin was heated using a heater patch matched with the sensor, maintaining the skin surface temperature at approximately 36 °C with a fluctuation of ±0.1 °C. Body posture changes can affect the transmission path of photons within the skin, introducing systematic bias and poor repeatability. Therefore, a posture-aiming method was used to stabilize forearm posture [8]. Two laser light sources were used to project pointers onto the forearm, where marks were drawn. Before each measurement, the forearm posture was adjusted to ensure that the laser pointers aligned with the marks on the forearm to restore the forearm posture to its initial state.

All subjects were fasting prior to the experiment. To avoid changes in skin moisture, the subjects wore the probe for 0.5 to 1 h in advance to ensure the stability of skin temperature and moisture, and then the subjects began to carry out OGTT formally. In addition, in order to avoid the subjects sweating during the OGTT, we not only ensure that the ambient temperature and humidity are stable and appropriate but also require the subjects to remain calm during the experiment. A waiting period of 0.5–1 h was required before the experiment to achieve thermal equilibrium between the skin and the heater patch. Each OGTT experiment was conducted within approximately 2 h. At the start of the OGTT, fasting blood glucose levels and diffuse reflectance intensity were recorded. Subsequently, the subjects consumed some food within a 5–10 min period. During the meal, the subjects ingested carbohydrates (about 60 g) and 50 mL of water. Thereafter, reference glucose and diffuse reflectance intensity values were measured simultaneously at intervals of about 10 min. Before data sampling, subjects were required to readjust to their initial posture with the aid of a laser alignment setup. Fingertip invasive blood glucose measurements were performed using two portable glucometers (GT-1820, Arkray, Shiga, Japan) as a reference. For detailed information on the specific apparatus and experimental procedures, further reference can be made to articles published by our team [8].

The detection outcomes from 25 individual OGTTs demonstrate a satisfactory correlation between the variations in differential absorbance and glucose levels. The average correlation coefficient was 0.79, with a maximum value of 0.89. Figure 3a,b presents one of the human experimental results of an individual OGTT at a wavelength of 1314 nm.

The average differential absorbance sensitivity from 25 individual OGTTs at $r_{A}$ = 1.7 mm and $r_{B}$ = 2.0 mm was $\frac{dA_{D}}{dC_{g}}$ of −0.00039 a.u.·(mmol/L)^−1^. Combining $\frac{dA_{D}}{d\mu_{s}}$ = 0.0102 a.u.·mm^−1^ with $\frac{dA_{D}}{dC_{g}}$ = 0.00039 a.u.·(mmol/L)^−1^, we derived $\frac{d\mu_{s}}{{dC}_{g}}$ = −0.038 mm^−1^ and $(d\mu_{s}/dC_{g})/\mu_{s}$ = −0.29% for 1314 nm. As described in Section 2.3, based on the Mie scattering theory, it is calculated that $(d\mu_{s}/dC_{g})/\mu_{s}$ is a constant in the range of 1000–1700 nm. Therefore, the $(d\mu_{s}/dC_{g})/\mu_{s}$ at any wavelength in the range of 1000–1700 nm is equal to −0.29% obtained at 1314 nm. Multiply $(d\mu_{s}/dC_{g})/\mu_{s}$ by the value of $\mu_{s}$ in the range of 1000–1700 nm [20] to obtain $d\mu_{s}/dC_{g}$ in the 1000–1700 nm range, as shown in Figure 3c. The $\frac{d\mu_{s}}{dC_{g}}$ for 2% intralipid is also shown and compared with the value in the human body, which is about one-fourth of that amount, where the $\frac{d\mu_{s}}{dC_{g}}$ of 2% intralipid was extrapolated from the study of Kohl et al. and Tamara [13,26].

## 3. Calculated Results of Sensitivity

Glucose absorbance sensitivity, denoted as $S_{A}=\frac{\partial A}{\partial \mu_{a}}\cdot \frac{d\mu_{a}}{{dC}_{g}}+\frac{\partial A}{\partial \mu_{s}}\cdot \frac{d\mu_{s}}{{dC}_{g}}$, depends on four factors: $\frac{d\mu_{a}}{{dC}_{g}}$ and $\frac{d\mu_{s}}{dC_{g}}$, and the resultant absorbance changes due to the absorption and scattering coefficient shifts. In this equation, $\frac{d\mu_{s}}{dC_{g}}$ was established in Section 2.4 Human Subject Experiments, while $\frac{d\mu_{a}}{{dC}_{g}}$ can use the reported data [27], as shown in Figure 4.

$\frac{\partial A}{\partial \mu_{a}}$ and $\frac{\partial A}{\partial \mu_{s}}$ are obtained through MC simulations. To calculate $\frac{\partial A}{\partial \mu_{a}}$, the scattering coefficient at each wavelength is kept constant. The MC simulation yields the absorbance when the absorption coefficient rises by 1 mm^−1^, from which $\frac{\partial A}{\partial \mu_{a}}$ is determined, as shown in Figure 5a. Similarly, to calculate $\frac{\partial A}{\partial \mu_{s}}$, the absorption coefficient at each wavelength is kept constant. The absorbance when the scattering coefficient rises by 1 mm^−1^ is calculated to obtain $\frac{\partial A}{\partial \mu_{s}}$, as shown in Figure 5b.

Based on $\frac{d\mu_{a}}{{dC}_{g}}$ and $\frac{\partial A}{\partial \mu_{a}}$ depicted in Figure 4 and Figure 5a, we can calculate the absorbance sensitivity due to glucose’s absorption, $S_{\mu_{a}}=\frac{\partial A}{\partial \mu_{a}}\cdot \frac{d\mu_{a}}{{dC}_{g}}$, as illustrated in Figure 5c. Similarly, based on $\frac{d\mu_{s}}{dC_{g}}$ and $\frac{\partial A}{\partial \mu_{s}}$ presented in Figure 3c and Figure 5b, we can calculate the absorbance sensitivity due to glucose’s scattering, $S_{\mu_{s}}=\frac{\partial A}{\partial \mu_{s}}\cdot \frac{d\mu_{s}}{{dC}_{g}}$, as demonstrated in Figure 5d.

From Figure 5c, it can be observed that $S_{\mu_{a}}$ is relatively small in the 1000–1400 nm range, with values approaching 0. However, in the 1400–1700 nm range, it is significantly larger, varying between 0.0004 a.u. and 0.0012 a.u., and is dependent on the SDSs. Within the used SDS range (1.7–2.9 mm), the larger the SDS, the greater the $S_{\mu_{a}}$. Figure 5d reveals that the $S_{\mu_{s}}$ varies noticeably in the 1350–1600 nm range with SDS (1.7–2.9 mm), reaching its maximum absolute value at SDS = 2.9 mm. Additionally, for SDS > 2.3 mm in the wavelength range where tissue absorption is stronger (1350–1700 nm), $S_{\mu_{s}}$ and $S_{\mu_{a}}$ have opposite signs, implying that these two effects partially offset each other’s sensitivity. Overall, by comparing Figure 5c,d, it is evident that the impact of the scattering effect, $S_{\mu_{s}}$, is greater than that of the absorption effect. Therefore, changes in tissue scattering due to glucose are the primary contributors to glucose sensitivity in the human body.

Figure 5e presents the sensitivity curves of the combined effects of glucose absorption and scattering on the human body at various SDSs across different wavelengths. It is observable that in the 1000–1350 nm range, there is little variation in sensitivity across wavelengths, with higher sensitivity achievable at the closer SDS of 1.7 mm. The sensitivity in this range is primarily determined by the scattering effect, $S_{\mu_{s}}$. In the 1350–1700 nm range, the sensitivity is jointly determined by the effects of $S_{\mu_{a}}$ and $S_{\mu_{s}}$. At the closer SDS of 1.7 mm, higher ‘positive’ sensitivity can be obtained, while at the farther SDS of 2.9 mm, higher absolute ‘negative’ sensitivity can be achieved.

In the practical application of human measurement, multiple wavelengths and multivariate analysis can be employed, considering the influence of fluctuations in other tissue components. The selected wavelengths should be not only sensitive to changes in glucose but also to hemoglobin, water, albumin, urea, and lactic acid, enabling multivariate analysis to effectively extract the blood glucose signal. According to Figure 5e, glucose sensitivity is high within the range of 1350–1700 nm. However, given the strong absorption of water near 1450 nm, the range of 1500 to 1700 nm is deemed more suitable for glucose sensing. The choice of measurement wavelength for other major blood components, such as albumin, hemoglobin, water, urea, and lactic acid, can refer to reference [27,28]. For instance, Ren et al. examined the absorption spectra of glucose, lactic acid, and urea in the near-infrared band, while Deng et al. obtained the absorption spectra of water. Similarly, the strong absorption peak of water around 1450 nm should be circumvented when sensing these components.

## 4. Results of Human Validation Experiment

To validate the estimated $\frac{d\mu_{s}}{{dC}_{g}}$ of the 1000–1700 nm range and the simulated absorbance sensitivity $S_{A}$, 25 human OGTTs were conducted. The used wavelengths are 1050 nm, 1219 nm, 1314 nm, 1380 nm, 1550 nm, and 1609 nm. The 25 human OGTT cases demonstrated a good correlation between $A_{D}$ and $C_{g}$ and were considered reliable experimental results for the validation of $\frac{d\mu_{s}}{{dC}_{g}}$ and sensitivity.

The differential absorbance $A_{D}$ at the SDSs of 1.7 mm and 2.0 mm was recorded; Figure 6a–d shows the human experiment results of 4 volunteers among 25 human OGTT cases. The sensitivity results for all 25 subjects are shown in Figure 6e. The average sensitivity $\frac{dA_{D}}{dC_{g}}$ of 25 cases is shown in Figure 6f and compared with the MC results.

As shown in Figure 6e, for the sensitivity results of each of the 25 subjects, the 1550 nm light exhibited the maximal sensitivity among the six wavelengths, with an average sensitivity of −0.00107 a.u.·(mmol/L)⁻^1^. As shown in Figure 6f, it can be observed that at the six wavelengths, the calculated and experimental sensitivity of differential absorbance *A*_D_ are close to each other. The relative errors between the average value of experimental sensitivity and the calculated sensitivity at 1050 nm, 1219 nm, 1314 nm, 1380 nm, 1550 nm, and 1609 nm are, respectively, 13%, 26%, 0%, 21%, 3%, and 9%. This indicates that the glucose signal sensitivity calculated in this study shows good consistency with the sensitivity acquired in the human experiment. Moreover, the sensitivity measured at 1550 nm closely matches the calculated sensitivity, indicating that the $\frac{d\mu_{s}}{{dC}_{g}}$ obtained at 1314 nm can be extended to wavelengths where human tissue absorption is stronger.

The measured sensitivity in the trial at 1219 nm shows a greater difference from the calculated sensitivity, primarily due to the weaker response of glucose changes at this wavelength, which is more susceptible to the influence of fatty substances in tissue. The 1380 nm light has strong absorption due to water, and in measurements, it is easily affected by changes in body tissue water content, which is a major reason for the discrepancy between the actual values and the calculated sensitivity.

## 5. Discussion

In the human experiment, it was found that using light at 1314 nm, the relative change in the scattering coefficient due to 1 mmol/L blood glucose concentration change was −0.29%. Based on these data and the Mie scattering theory, we assume that the relative change in the scattering coefficient due to glucose variation is −0.29% for all lights within the 1000–1700 nm range, which are used for the calculation of the sensitivity of the absorbance. The human OGTTs were conducted to validate the calculated sensitivity results of glucose signals.

In our measurements on 25 subjects using 1314 nm, the relative change rate of scattering coefficient with blood glucose concentration $(d\mu_{s}/dC_{g})/\mu_{s}$ was estimated to be −0.29% ± 0.098%. The variation among subjects is likely attributable to differences in skin structure, tissue components, and the physiological state of the skin.

The $(d\mu_{s}/dC_{g})/\mu_{s}$ obtained in this paper is compared with the published results, as shown in Table 1.

It can be seen from Table 1 that the $(d\mu_{s}/dC_{g})/\mu_{s}$ measured in vivo in the reported studies is much greater than that in the phantom (such as intralipid), which is about 11–306 times that in 2% intralipid. The result of this paper is a factor of 14.5, which is also in this range and is on the small side.

The results measured in vivo in Table 1 were mostly obtained by the OCT method, and there was a significant difference between their values. The reason for this significant difference may be due to the different measurement objects, measurement depths, and wavelengths used in various studies. In addition, the research of GAO et al. provided an explanation, that is, they found that the scattering coefficient change with glucose in blood and interstitial fluid is opposite [29]. The $(d\mu_{s}/dC_{g})/\mu_{s}$ in blood vessels with a diameter of more than 20 µm is +2.7%, while the $(d\mu_{s}/dC_{g})/\mu_{s}$ is −0.75% in the interstitial fluid of the dermis. Therefore, different blood and interstitial fluid distributions in tissues will lead to different measurement results. In Table 1, the other studies using OCT measurements, except GAO, only give the results of $(d\mu_{s}/dC_{g})/\mu_{s}$ in a specific depth range without distinguishing between blood and interstitial fluid. Therefore, the different tissue objects with different probing depths can significantly affect the magnitude of $(d\mu_{s}/dC_{g})/\mu_{s}$.

In this paper, we used near-infrared diffuse reflectance spectroscopy. The measurement principle is different from the OCT method. It does not locate at a specific depth in detection. The measured diffuse light passes through all layers of skin tissue, including the epidermis, dermis, and a part of subcutaneous tissue. The average tissue scattering change caused by glucose in the whole light path will affect the results, so there is a certain difference with the OCT results.

### 5.1. Comparison of Calculated Sensitivity with Experimental Sensitivity

The MC method demonstrates the sensitivity of absorbance caused by glucose for the wavelengths within the 1000–1700 nm range. Then, the MC results were well validated by the human experiment at six typical wavelengths within this range. The experimental results of the differential absorbance of SDSs of 1.7 mm and 2.0 mm show good consistency with the theoretical sensitivity calculated by the MC method. Additionally, we observed that among these six wavelengths, the sensitivity of differential absorbance is slightly higher for the three longer wavelengths (1380 nm, 1550 nm, and 1609 nm) and slightly lower for the three shorter wavelengths (1050 nm, 1219 nm, and 1314 nm). Therefore, for the differential measurement method, longer wavelengths should be preferred while avoiding wavelengths with strong water absorption, such as the 1500–1600 nm range.

### 5.2. Reasonably Choosing SDSs Based on Sensitivity Results

Our method can determine the absorbance sensitivity caused by glucose, $\frac{dA}{{dC}_{g}}$, at different SDSs for the wavelengths within the 1000–1700 nm range, serving as a reference for selecting reasonable SDS. Based on the calculated result, we can identify the “floating reference point” on the human body [30], i.e., the SDS where absorbance sensitivity is close to zero, as shown in Figure 5e. When using light within the 1000–1300 nm range, the sensitivity is close to zero at SDS = 2.6 mm. The detection signal at this SDS only varies with the human body background and can serve as a reference for eliminating physiological background interference. When using light within the 1300–1560 nm range, the “floating reference point” is about 1.7–2.0 mm; for light within the 1560–1700 nm range, it is within the 2.6–2.9 mm range.

The choice of SDS according to the calculated sensitivity also dictates the requirements for the instrument noise level. To achieve higher sensitivity levels, a larger SDS should be chosen, but the larger the SDS, the weaker the detected light energy. In this case, it is necessary to minimize instrument noise. For instance, when using light within the 1500–1560 nm range, the sensitivity at SDS = 2.9 mm is −0.00083 a.u. To discern a 1 mmol/L glucose concentration change, the random noise of the absorbance at this SDS needs to be controlled to 0.00061 a.u. or less. If the instrument’s performance cannot meet this requirement, then the measurements should use a smaller SDS to ensure accuracy.

### 5.3. Comparison of Glucose Sensitivity in Human Body and Intralipid Solution

The methodology presented in this paper can also be applied to estimate the sensitivity of glucose measurement in intralipid phantoms. Figure 7a shows the absorbance sensitivity for detecting glucose in intralipid solutions, while Figure 7b shows the absorbance sensitivity due to glucose’s absorption change, and Figure 7c shows that due to glucose’s scattering change. Comparing Figure 5e and Figure 7a, it is evident that the glucose detection sensitivity in the human body is greater than that in intralipid. The primary reason is that in the human body, the change in the scattering coefficient caused by glucose is greater than in intralipid, as shown in Figure 3c. Moreover, by comparing Figure 5c,d, we can understand that the glucose detection sensitivity in the human body is mainly contributed by a glucose-caused scattering change. However, the situation is different in the intralipid solutions; as observed in Figure 7b,c, the effects of absorption and scattering changes on sensitivity in the medium are similar in magnitude but opposite in direction, which counterbalance each other. Consequently, the glucose sensitivity in the intralipid solution is lower than in the human body. In short, despite the absorption changes induced by glucose contributing similarly to sensitivity in both intralipid and human tissues, the scattering variations caused by glucose are markedly different in these two media. This distinction is the primary factor leading to the differences in glucose sensitivity observed between intralipid and human tissues.

### 5.4. Preliminary Research on Diversity in Sensitivity among Individuals

Through experimentation, we learned that glucose detection sensitivity is influenced by the optical properties of human tissue. Differences in skin’s optical properties among individuals directly determine the level of sensitivity. In our study, we measured the sensitivity of differential absorbance for the forearm skin in 25 subjects and found their difference due to skin characteristics. It needs to be clarified that the discussion on the diversity in sensitivity among individuals is preliminary, as it is only for the tests on the forearm (the extensor side) of these Asian subjects.

Skin’s optical properties vary with age and gender. In the experiment, we divided the 25 cases into four groups based on the subject gender and age: elderly male group (6 individuals), elderly female group (7 individuals), young male group (6 individuals), and young female group (6 individuals), and we calculated the average sensitivity for each group, as shown in Figure 8.

The results indicate that at wavelengths of 1550 nm and 1609 nm, the amplitude of glucose detection sensitivity is larger with less measurement interference, so particular attention should be paid to results at these wavelengths. In Figure 8a, we can see that the glucose sensitivity in females is slightly higher than in males. However, Figure 8b shows no significant difference in sensitivity between elderly subjects (over 65 years) and younger subjects (under 35 years).

According to research by Kono et al. [31], gender does not significantly affect the skin’s absorption coefficient, but there is a notable gender difference in the scattering coefficient when using light in the 1000–1600 nm range, with the average scattering coefficient in female skin being about 18.8% higher than in male skin. Our study found that due to the higher scattering coefficient, their $\frac{d\mu_{s}}{{dC}_{g}}$ is larger, and the change in the scattering coefficient has a greater impact on the sensitivity of glucose signals. This might explain the slightly higher glucose detection sensitivity in females.

We also noted that skin differences among people of different ages are mainly manifested in water content. According to research by Nakagawa et al. [32], the water content in the dermis of the elderly (73.3 ± 1.6%) is higher, while in younger people, it is slightly lower at 69.3 ± 1.7%. Although elderly people might have a higher absorption coefficient due to increased skin water content, our experiments did not observe a significant impact of age-related differences in the absorption coefficient on the sensitivity, as shown in Figure 8b. However, elderly females, compared to the other three groups, have higher absorption and scattering coefficients. These combined effects could lead to slightly higher glucose signals in this group, as shown in Figure 8c.

## 6. Conclusions

This study estimated the scattering coefficient change caused by glucose in human skins, yielding values for glucose signal sensitivity at the wavelengths in the 1000–1700 nm range and at the SDSs in the 1.7–2.9 mm range. By comparing the sensitivity results of differential absorbance obtained from human experiments with calculated results by the MC method, we validated the scattering coefficient change caused by glucose on the waveband of 1000–1700 nm, which is extended from the result of 1314 nm and reaches to −0.29%. Based on the sensitivity results, we identified the zero points of blood glucose detection sensitivity on the human body—the ‘floating reference positions’—as well as the signal-to-noise requirements for the measurement system. We also explored the differences and reasons behind the sensitivity of glucose measurements in human bodies and phantoms. And we compared and discussed the differences in sensitivity due to skin characteristics that varied with age and gender and their potential reasons.

The findings of this paper present the sensitivity of near-infrared diffuse reflectance spectroscopy for in vivo blood glucose signal measurement, considering both the dimensions of different wavelengths and various source–detector distances. This provides a crucial reference for understanding the characteristics of blood glucose signals within the human body. It facilitates the identification of glucose signals amidst the complex physiological background interferences encountered in in vivo measurements.

## Figures and Tables

**Figure 1 sensors-24-05879-f001:**
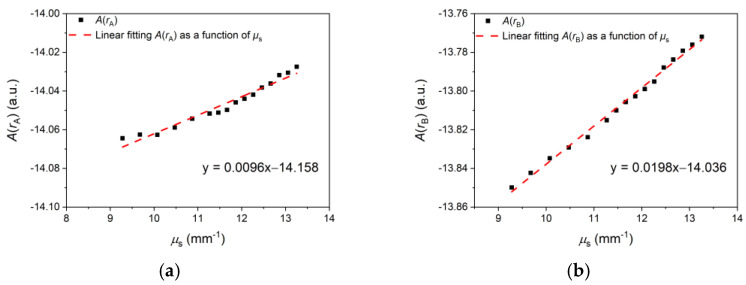
(**a**) Linear fitting curve of $A(r_{A})$ as a function of $\mu_{s}$ at 1314 nm. (**b**) Linear fitting curve of $A(r_{B})$ as a function of $\mu_{s}$ at 1314 nm.

**Figure 2 sensors-24-05879-f002:**
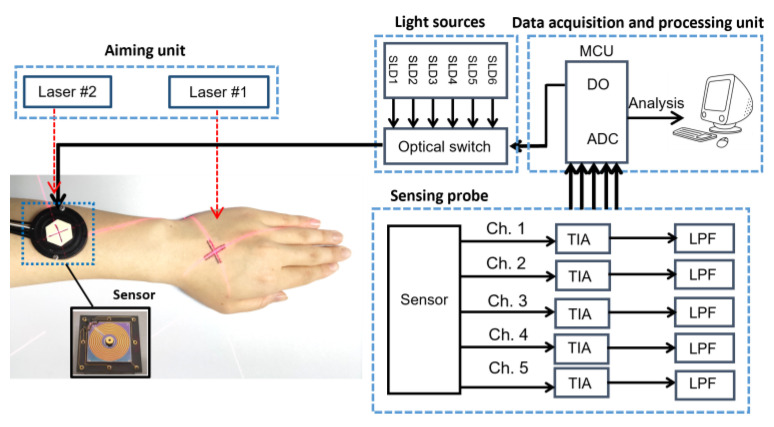
Block diagram of the experiment system [8].

**Figure 3 sensors-24-05879-f003:**
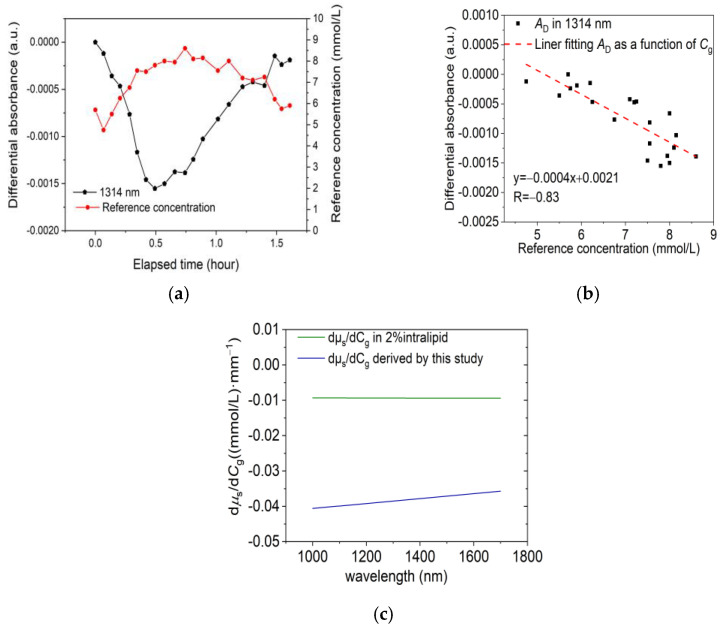
(**a**) The differential absorbance (A_D_) of the 1314 nm and the glucose reference value (Cg). (**b**) The fitted line of A_D_ and Cg, with the correlation coefficient R between them. (**c**) Scattering coefficient variation per unit glucose concentration change $\frac{d\mu_{s}}{dC_{g}}$ in the 1000–1700 nm range.

**Figure 4 sensors-24-05879-f004:**
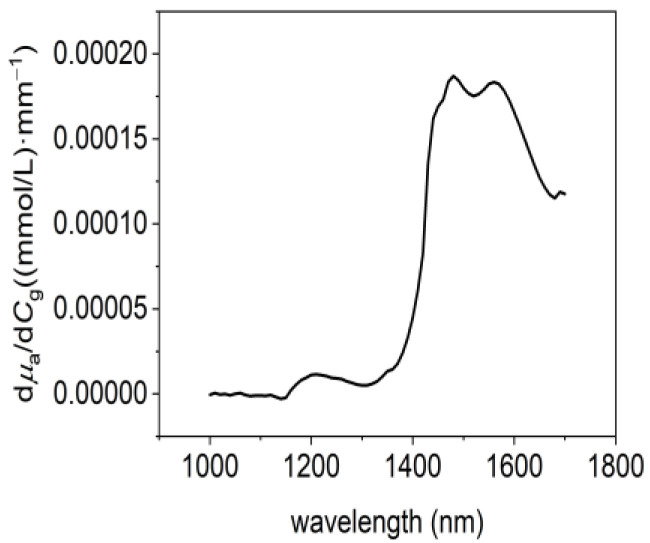
Variation in absorption coefficient due to per unit glucose concentration change $\frac{d\mu_{a}}{dC_{g}}$.

**Figure 5 sensors-24-05879-f005:**
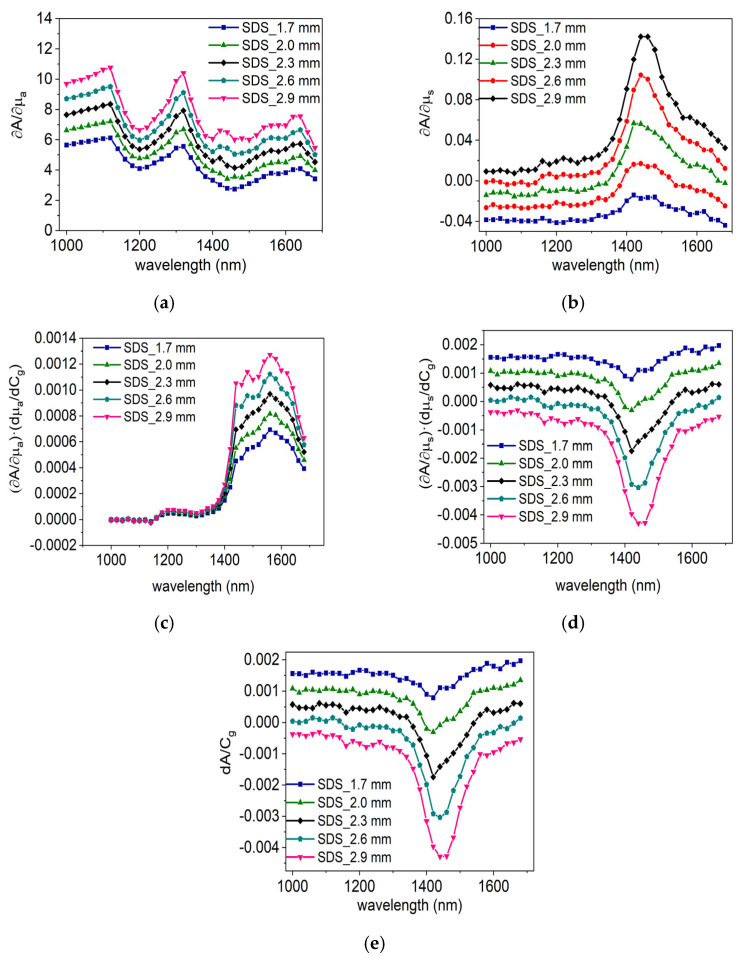
(**a**) $\frac{\partial A}{\partial \mu_{a}}$ obtained by MC simulation, and (**b**) $\frac{\partial A}{\partial \mu_{s}}$ obtained by MC simulation. (**c**) The absorbance sensitivity due to glucose’s absorption change, (**d**) the absorbance sensitivity due to glucose’s scattering change, and (**e**) the absorbance sensitivity at five SDSs from the calculation results.

**Figure 6 sensors-24-05879-f006:**
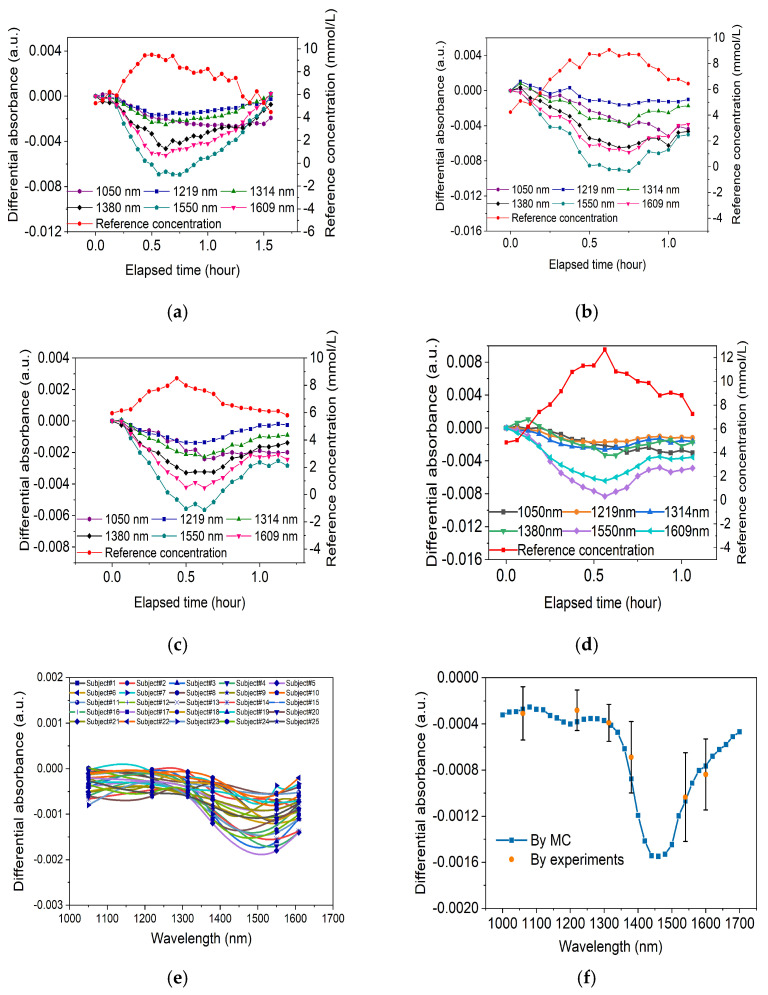
Results of human experiments compared with the calculated results. (**a**–**d**) The change in differential absorbance $A_{D}$ from 4 volunteers at the six wavelengths and the glucose reference value (Cg); (**e**) the experimental sensitivity $\frac{dA_{D}}{dC_{g}}$ of all 25 cases; (**f**) calculated and average experimental sensitivity $\frac{dA_{D}}{dC_{g}}$ of 25 cases.

**Figure 7 sensors-24-05879-f007:**
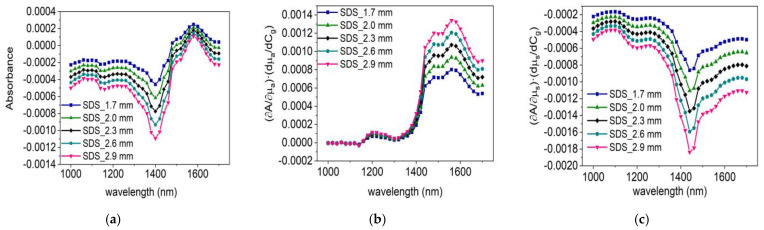
Results of sensitivity calculations in intralipid solution: (**a**) variation of sensitivity with wavelength at different source–detector separations; (**b**) impact of glucose–induced absorption changes on sensitivity; (**c**) impact of glucose-induced scattering changes on sensitivity.

**Figure 8 sensors-24-05879-f008:**
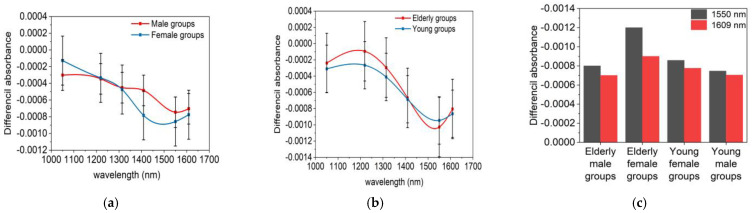
(**a**) Glucose detection sensitivity of differential absorbance for male and female skin; (**b**) glucose detection sensitivity of differential absorbance for elderly and young skin; and (**c**) glucose detection sensitivity of differential absorbance among the four groups of subjects at wavelengths of 1550 nm and 1609 nm.

**Table 1 sensors-24-05879-t001:** Comparison of the relative change rate of scattering coefficients with blood glucose concentration measured in vivo and in a phantom.

Optical Method	NIR Diffuse Reflectance	NIR Diffuse Reflectance	OCT						
Source of data	Kohl, M. et al. [13]	This study	Larin, K. et al. [12]	Kohl, M. et al. [15]	Bruulsema, J.T. et al. [14]	Larin, K.V. et al. [16]		GAO et al. [29]	
Wavelength	700 nm	1314 nm	1300 nm	910 nm	650 nm	1300 nm		1300 nm	
Source of sample	Tissue-simulating phantoms	Human left forearm	New Zealand rabbits	Mouse	Human abdomen	Human left forearm		The junction of the fingers of the human body	
Detecting depth/position	-	-	260~400 μm	100~300 μm	-	550~600 μm	380~500 μm	Blood	Interstitial fluid
(dμs/dCg)/μs	−0.02%	−0.29%	−0.22%	−2.70%	−0.11~−0.34%	−6.12%	−2.70%	2.70%	−0.75%
The ratio of (dμs/dCg)/μs in vivo to that in phantoms	-	14.5	11	135	50~170	306	135	−135	37.5

## Data Availability

The original contributions presented in the study are included in the article, further inquiries can be directed to the corresponding authors.

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
