# Peer review of "Evaluation and Validation on Sensitivity of Near-Infrared Diffuse Reflectance in Non-Invasive Human Blood Glucose Measurement"

_sensors, 2024, doi:10.3390/s24185879_

Round 1
Reviewer 1 Report
Comments and Suggestions for Authors
The authors propose an interesting paper investigating optical absorption and scattering changes related to blood sugar content. The authors didn’t design a new sensing scheme but utilized an already published one based on NIR diffuse reflectance (Han et al. Applied Spectroscopy 2022, 76, 1100) and conducted in-vivo human experimentation with this sensing scheme.
The proposed paper mostly focus on changes in the scattering coefficient due to blood glucose during 25 human OGTTs. This study thus provides further data to the scientific community that could help designing NIR based blood glucose sensing system in the future.
Nevertheless, the proposed paper could be improved if the authors include a deeper comparison of their results with the many published results on NIR in-vivo scattering:
A) In the introduction, the authors cite previous works on scattering coefficient change with in-vivo blood glucose (e.g. ref. 12, 13). Since this is a focus of the proposed paper, the author should compare (in the section “Discussion”) their own results on scattering coefficient change more deeply with the ones already published.
Other small remarks:
B.1) Line 118: “we introduce a method that integrates MC simulations with in vivo experiments”. Line 113: “Additionally, we conducted MC simulations for a three-layer skin model to acquire the absorbance sensitivity of glucose in this wavelength range.” A detailed description of the utilized MC simulations (e.g. algorithm, exact geometry and parameters, and this also for Fig.1.) should be given in Section 2 “Methods and experiments” (a bit like on lines 265-269).
B.2) Even if the experimental setup can be found in ref. 8. and a description is given in lines 209-219, a figure could help the reading. Is it Fig.2? (Fig. 2 is missing on the downloaded .pdf)
B.3) In the caption of fig.3c, the origin of the data should be given.
Author Response
|
Comments A: In the introduction, the authors cite previous works on scattering coefficient change with in-vivo blood glucose (e.g. ref. 12, 13). Since this is a focus of the proposed paper, the author should compare (in the section “Discussion”) their own results on scattering coefficient change more deeply with the ones already published. |
|
Response A: As mentioned by the reviewers, scattering coefficient change with in vivo blood glucose is a major concern of this paper. Therefore, as suggested by the reviewers, we have added Table 1 in the Discussion section, which compares the relative change rate of scattering coefficients with blood glucose obtained in this paper with published results. In the revised manuscript's Discussion section, line -366-, we added the following discussion: “In our measurements on 25 subjects using 1314 nm, the relative change rate of scattering coefficient with blood glucose concentration (dμs/dCg)/μs was estimated to be -0.29% ± 0.098%. The variation among subjects is likely attributable to differences in skin structure, tissue components, and the physiological state of the skin. The (dμs/dCg)/μs obtained in this paper is compared with the published results, as shown in Table 1. Table 1. Comparison of the relative change rate of scattering coefficients with blood glucose concentration measured in vivo and in a phantom.
It can be seen from Table 1 that the (dμs/dCg)/μs measured in vivo in the reported is much greater than that in the phantom (such as intralipid), which is about 11-306 times of that in 2% intralipid. The result of this paper is a factor of 14.5, which is also in this range and is on the small side. The results measured in vivo in Table 1 were mostly obtained by the OCT method, and there was a significant difference between their values. The reason for this significant difference may be due to the different measurement objects, measurement depths, and wavelengths used in various studies. In addition, the research of GAO et al. provided an explanation, that is, they found that the scattering coefficient change with glucose in blood and interstitial fluid is opposite. The (dμs/dCg)/μs in blood vessels with a diameter of more than 20 μm is +2.7%, while the (dμs/dCg)/μs is -0.75% in interstitial fluid of dermis. Therefore, different blood and interstitial fluid distributions in tissues will lead to different measurement results. In Table 1, the other studies using OCT measurements, except GAO, only give the results of (dμs/dCg)/μs in a specific depth range without distinguishing between blood and interstitial fluid. Therefore, the different tissue objects with the different probing depths can significantly affect the magnitude of (dμs/dCg)/μs. In this paper, we used the near-infrared diffuse reflectance spectroscopy. The measurement principle is different from the OCT method. It does not locate at a specific depth in detection. The measured diffuse light passes through all layers of skin tissue, including epidermis, dermis and a part of subcutaneous tissue. The average tissue scattering change caused by glucose in the whole light path will affect the results, so there is a certain difference with the OCT results.” |
|
Comments B.1): Line 118: “we introduce a method that integrates MC simulations with in vivo experiments”. Line 113: “Additionally, we conducted MC simulations for a three-layer skin model to acquire the absorbance sensitivity of glucose in this wavelength range.” A detailed description of the utilized MC simulations (e.g. algorithm, exact geometry and parameters, and this also for Fig.1.) should be given in Section 2 “Methods and experiments” (a bit like on lines 265-269). |
|
Response B.1): Thanks for your suggestions. We have accordingly added a subsection "2.2 Monte Carlo Simulation" in the "Methods and experiments" section to describe the algorithm, geometric parameters, etc. of the MC simulation in detail on line -155- in the revise manuscript. In order to avoid repetition, the original description of the geometric parameters of the MC simulation in lines 265-269 has been deleted. The added description is as follow: “2.2 Monte Carlo Simulation The tetrahedron-based inhomogeneous Monte Carlo simulator MC eXtreme (MCX) [19], version 1.9.7 (v2022.10, Heroic Hexagon), proposed by Shen et al., is used in this paper. Simulation parameters are set as follows: The skin is modeled as a three-layer structure, with the epidermis, dermis, and subcutaneous layers having thicknesses of 0.1 mm, 1.0 mm, and 10.0 mm, respectively. The optical parameters for each skin layer are derived from literature [20]. The probe interface is made of glass with a refractive index of n = 1.4 and a thickness of 0.2 mm. Diffuse reflectance photons are collected at five SDSs of 1.7 mm, 2.0 mm, 2.3 mm, 2.6 mm, and 2.9 mm, with 1010 incident photons.” |
|
Comments B.2): Even if the experimental setup can be found in ref. 8. and a description is given in lines 209-219, a figure could help the reading. Is it Fig.2? (Fig. 2 is missing on the downloaded .pdf) |
|
Response B.2): In the previously submitted manuscript, Figure 2 was not displayed, possibly due to an upload problem. It has been added in the line -252- of the revised manuscript. The added figure 2. is as follow:
Figure 2. Block diagram of the experiment system [8]. |
|
Comments B.3): In the caption of fig.3c, the origin of the data should be given. |
|
Response B.3): The scattering coefficient change caused by the change of glucose concentration dμs/dCg in 2% intralipid solution in the 1000-1700 nm range in the manuscript was calculated according to the research results of Kohl M. et al. and Tamara et al. Kohl M. et al. confirmed that the relative change rate of scattering coefficient caused by glucose in the phantom is in line with the results calculated from Mie theory, but the study only gave the results at the wavelength of 700 nm and 950 nm. Therefore, the Mie scattering theory is used to calculate the dμs/dCg in 2% intralipid solution in the range of 1000-1700 nm. The parameters used are based on the research of Kohl M. et al. and Tamara et al. At the suggestion of the reviewer, we have added these two references to the revised manuscript and added the following in lines -267-: “, where the of 2% intralipid was extrapolated from the study of Kohl et al. and Tamara [26-27].” [26] Kohl, M.; Essenpreis, M.; Cope, M. The influence of glucose concentration upon the transport of light in tissue-simulating phantoms. Physics in Medicine & Biology. 1995, 40(7), 1267. [27] Troy, T.; T, S. Optical properties of human skin in the near infrared wavelength range of 1000 to 2200 nm. Journal of biomedical optics. 2001, 6(2), 167-176. |

Reviewer 2 Report
Comments and Suggestions for Authors
Please see the attached file.

Not Applicable.
Author Response
|
Comments 1: Line-176 – the statement “We selected 1314 nm light to measure d?s/d?g of human skin, less affected by glucose absorption, hemoglobin, water content, and body temperature fluctuations.” needs proper references. |
|
Response 1: As suggested by the reviewer, we have supplemented and annotated the reference for this sentence on the line -182- in the revised manuscript, as “We selected 1314 nm light to measure d?s/d?g of human skin, minimally affected by glucose absorption [22], hemoglobin [23], water content [24], and body temperature fluctuations [25].” The supplemented references are: [22] Tarumi, M.; Shimada, M.; Murakami, T.; Tamura, M.; Shimada, M.; Arimoto, H.; Yamada, Y. Simulation study of in vitro glucose measurement by NIR spectroscopy and a method of error reduction. Physics in Medicine & Biology. 2003, 48(15), 2373. https://doi.org/10.1088/0031-9155/48/15/309. [23] Kuenstner, J.; Norris, K. Spectrophotometry of human hemoglobin in the near infrared region from 1000 to 2500 nm. Journal of Near Infrared Spectroscopy. 1994, 2(2), 59-65. [24] Deng, R.; He, Y.; Qin, Y.; Chen, Q.; Chen, L. Measuring pure water absorption coefficient in the near-infrared spectrum (900--2500 nm). Journal of Remote Sensing. 2012, 16(1), 192-206. [25] Iorizzo, T.; Jermain, P.; Salomatina, E.; Muzikansky, A.; Yaroslavsky, A. Temperature induced changes in the optical proper-ties of skin in vivo. Scientific Reports. 2021, 11(1), 754. Below, I would like to provide a detailed explanation of the reasons for choosing 1314 nm light to measure d?s/d?g of human skin. It is because it is less affected by changes in glucose absorption, hemoglobin absorption, water absorption, and fluctuations in body temperature. Tarumi measured the absorption coefficient of solutions of different concentrations of glucose in the 800-1800 nm band, as shown in Figure 1 [22]. It can be seen that in the 1000-1700 nm range, the absorption of glucose is relative strong at 1400-1600 nm, but very weak at 1000-1300 nm. According to the results in the literature, d?a/d?g is calculated to be 1.48·10-5 mm -1/ (mmol/L) -1 at 1314 nm. In 1000-1300 nm, the optical path length is about 4-7 mm for 1.7-2.9 mm of the SDSs used in our study, and the absorbance change caused by 1 mmol/L glucose concentration is only about 5.29·10-5-10.36·10-5 a.u., so we think it is small enough to be ignored.
Figure 1. Changes in the spectra of the absorption coefficient, ?a(λ), of aqueous glucose solution with the increase in Cg from the spectrum of pure water. The glucose band around 1600 nm increases while the water band around 1400 nm decrease as Cg increases from 0 to 10000 mg·dl-1. In Kuenstner's study, the absorption coefficients of several types of hemoglobin were measured in the 1000-2500 nm band (shown in Figure 2) [23], where we focused on the absorption of oxygenated and deoxygenated hemoglobin. The two kind of hemoglobin have strong absorption in 600-1200 nm range, and have very weak absorption near 1314 nm that is less than 0.0001 mm-1/ (mmol/L) -1. In the optical path of 4-7 mm, the change of absorbance caused by the hemoglobin change of 1 mmol/L is only about 0.0004-0.0007 a.u. Therefore, the absorption change of hemoglobin at 1314 nm is considered as small when the concentration of hemoglobin does not change much in measurements.
Figure 2. The absorption spectrum of each of the hemoglobin species following subtraction of the spectrum of the appropriate background. A is carboxyhemoglobin, B is deoxyhemoglobin, C is oxyhemoglobin, and D is methemoglobin. Deng et al. measured the absorption coefficient of pure water in the 400-2500 nm range (shown in Figure 3) [24]. In the range of 1000-1700 nm, the absorption of water is the strongest near 1450 nm, while the absorption is very weak at 1314 nm, which is about 3·10-5 mm-1/ (mmol/L) -1. In the optical path of 4-7 mm, the change of absorbance caused by the change of 1 mmol/L water concentration is only about 0.00012-0.00021 a.u. Therefore, the change of water absorption at the wavelength of 1314 nm can be ignored when the water concentration does not change dramatically during the measurement.
Figure 3. Unioned absorption coefficients of pure water between 400 nm and 2500 nm Iorizzo measured the tissue optical parameters of mouse skin in vivo, including absorption coefficient μa, scattering coefficient μs and anisotropy factor g, in the 400-1600 nm range at 25 ℃, 36 ℃ and 60 ℃, respectively (as shown in Figure 4(a-c)) [25]. The results show that the μa, μs and g at 1314 nm change little when the temperature changes from 25 ℃ to 36 ℃. In addition,the skin temperature is controlled at 36 ± 0.1 ℃ in our experimental measurement, so the effect of temperature can be ignored at 1314 nm.
Figure 4. (a) Absorption coefficients between 400–1650 nm. (b) Scattering coefcients between 400–1650 nm. (c) Anisotropy factors between 400–1650 nm. Dashes—25 °C. Dashes and dots—36 °C. Solid line—60 °C. Bars—standard errors. Overall, the light signal at 1314 nm is not sensitive to the absorption of glucose, water and hemoglobin, nor to the change of temperature. Therefore, under relatively stable experimental conditions, 1314 nm is the preferred wavelength for observing the changes in tissue scattering coefficient caused by glucose. |
|
Comments 2: As the block diagram of the Fig. 2 is missing, it is quite hard to follow. |
|
Response 2: In the previously submitted manuscript, the block diagram of Figure 2 could not be displayed due to an upload error. It has been added on the line -252- in the revised manuscript. |
|
Comments 3: I would suggest adding an experimental setup with the block diagram in Fig. 2. Although the setup was described in Ref. [8], however, I believe this will help to understand this work effectively. |
|
Response 3: On the line- 252- in the revised manuscript, we supplemented the block diagram of figure 2. The added figure 2. is as follow:
Figure 2. Block diagram of the experiment system [8]. |
|
Comments 4: For collecting the data from human subjects, did the author monitor skin moisture? Is there any effect of the Humidity on the measurement? |
|
Response 4: According to our research, the state of skin moisture is mainly affected by factors such as environmental temperature, humidity, skin temperature, and the subject's emotions. Changes in skin moisture can affect spectral data. In order to maintain a stable skin moisture during the blood glucose measurement, the measures we take are: (1) The subject wears the probe for 0.5 to 1 hour in advance to ensure the stability of the skin temperature and moisture before the subject starts the formal OGTT. (2) Ensure that the temperature and humidity of the experimental environment are stable and suitable, and require the subject to maintain emotional calm during the experiment to avoid obvious sweating in the subject during the OGTT. To further illustrate that skin moisture remained stable after the start of the OGTT, we collected the change in differential absorbance in subjects with stable fasting glucose from differential absorbance at the initial time, as shown in the following figure 5:
Figure 5. Change in differential absorbance in subjects with stable fasting glucose from differential absorbance at the initial time. It can be seen that after wearing the probe, the signal drifts and stabilizes after about 1 hour. In the following 1~1.5 hour, the differential absorbance remained stable, and the change of differential absorbance at 1550 nm wavelength was only ± 0.05%, which indicated that the skin moisture of the subjects could remain stable during this period. In line -234- of the revised manuscript, we added: "To avoid changes in skin moisture,the subjects wore the probe for 0.5 to 1 hour in advance to ensure the stability of skin temperature and moisture, and then the subjects began to carry out OGTT formally. In addition, in order to avoid the subjects sweating during the OGTT, we not only ensure that the ambient temperature and humidity are stable and appropriate, but also require the subjects to remain calm during the experiment.” |
|
Comments 5: Line--251--“And then -0.29% of 1314 nm can be extended to all wavelengths in 1000 - 1700 nm range, as shown in Figure 3(c).”- How the measured data at 1314 nm was extended to other wavelengths? |
|
Response 5: Our extended method is based on equation (6). It is deduced from equation (6) that (dμs/dCg)/μs is approximately equal at each wavelength in the 1000-1700 nm range. Therefore, the (dμs/dCg)/μs value at 1314 nm is used directly for the (dμs/dCg)/μs values at other wavelengths. In line -261- of the text, we have changed this sentence to make it clearer: “As described in Section 2.3, based on Mie scattering theory, it is calculated that (dμs/dCg)/μs is a constant in the range of 1000-1700 nm. Therefore, the (dμs/dCg)/μs at any wavelength in the range of 1000-1700 nm is equal to -0.29% obtained in 1314 nm. Multiply (dμs/dCg)/μs by the value of μs in the range of 1000-1700 nm [20] to obtain dμs/dCg in the 1000-1700 nm range, as shown in Figure 3 (C).” |
|
Comments 6: Line- 301—“ To avoid the absorption peaks of water, the wavelength range of 1500 – 1700 nm can be selected for detection.” The water shows most of the background signal from the human skin. However, there is a considerable effect of Albumin as a background component in the NIR region. Did the author consider the effect of other interfering blood components like Albumin, Urea, and Lactate. |
|
Response 6: As mentioned by the reviewer, the background components will have a great impact on the measurement, in which water is an important factor, and other factors such as hemoglobin, albumin, urea and lactic acid cannot be ignored. During the 2-3 hours of OGTT, the contents of albumin, urea and lactic acid are relatively stable and have little influence on the measurement, but their influence cannot be ignored when the measurement is carried out in vivo for a long-time monitoring. We can perform a multivariate analysis using the data at multiple wavelengths to remove the interference of these components. In multivariate analysis, the selected wavelengths should include not only the wavelength with high glucose sensitivity, but also the wavelengths sensitive to the changes of hemoglobin, water, albumin, urea and lactic acid. When choosing the wavelengths for these components, it is also necessary to avoid the strong absorption band of water near 1450 nm. Therefore, in line -312- of the revised manuscript, we have revised “To avoid the absorption peaks of water, the wavelength range of 1500–1700 nm can be selected for detection." to blow: “In practical application of human measurement, multiple wavelengths and multivariate analysis can be employed, as considering the influence of fluctuations in other tissue components. The selected wavelengths should not only be sensitive to changes in glucose but also to hemoglobin, water, albumin, urea and lactic acid, enabling multivariate analysis to effectively extract the blood glucose signal. According to Figure 5(e), glucose sensitivity is high within the range of 1350 - 1700 nm, However, given the strong absorption of water near 1450 nm, the range of 1500 to 1700 nm is deemed more suitable for glucose sensing. The choice of measurement wavelength for other major blood components such as albumin, hemoglobin, water, urea and lactic acid can refer to reference [29-30]. For instance, Ren examined the absorption spectra of glucose, lactic acid and urea in the near infrared band, while Deng et al obtained the absorption spectra of water. Similarly, the strong absorption peak of water around 1450 nm should be circumvented when sensing these components.” |

Round 2
Reviewer 2 Report
Comments and Suggestions for Authors
The authors addressed all the reviewer's comments. As they clarified and addressed all the points in the updated manuscript, this work can be published.